# Circulating Tumor DNA for Prediction of Complete Pathological Response to Neoadjuvant Radiochemotherapy in Locally Advanced Rectal Cancer (NEORECT Trial)

**DOI:** 10.3390/cancers16244173

**Published:** 2024-12-14

**Authors:** Tatiana Mögele, Michael Höck, Florian Sommer, Lena Friedrich, Sebastian Sommer, Maximilian Schmutz, Amadeus Altenburger, Helmut Messmann, Matthias Anthuber, Thomas Kröncke, Georg Stüben, Martin Trepel, Bruno Märkl, Sebastian Dintner, Rainer Claus

**Affiliations:** 1Pathology, Faculty of Medicine, University of Augsburg, 86156 Augsburg, Germany; tatiana.moegele@uk-augsburg.de (T.M.); bruno.maerkl@uk-augsburg.de (B.M.); 2Bavarian Cancer Research Center (BZKF), Germany; maximilian.schmutz@uk-augsburg.de (M.S.); martin.trepel@uk-augsburg.de (M.T.); 3Radiotherapy, Faculty of Medicine, University of Augsburg, 86156 Augsburg, Germanygeorg.stueben@uk-augsburg.de (G.S.); 4General and Visceral Surgery, Faculty of Medicine, University of Augsburg, 86156 Augsburg, Germany; florian.sommer@uk-augsburg.de (F.S.); matthias.anthuber@uk-augsburg.de (M.A.); 5Diagnostic and Interventional Neuroradiology, Faculty of Medicine, University of Augsburg, 86156 Augsburg, Germany; lena.friedrich@uk-augsburg.de (L.F.); thomas.kroencke@uk-augsburg.de (T.K.); 6Hematology and Oncology, Faculty of Medicine, University of Augsburg, 86156 Augsburg, Germany; 7Gastroenterology, Faculty of Medicine, University of Augsburg, 86156 Augsburg, Germany; helmut.messmann@uk-augsburg.de; 8Comprehensive Cancer Center Augsburg (CCCA), 86156 Augsburg, Germany

**Keywords:** rectal cancer, neoadjuvant radiochemotherapy, complete pathological response, circulating tumor DNA, response prediction

## Abstract

The NEORECT trial explored the use of circulating tumor DNA (ctDNA) dynamics as a biomarker to monitor treatment response in rectal cancer patients undergoing nCRT. Although ctDNA patterns showed some correlation with treatment outcomes, specificity was low, indicating limitations in using ctDNA alone to predict pathologic remission. The findings highlight the potential and challenges of ctDNA-based monitoring, suggesting that broader genetic analysis and larger studies are needed to enhance precision in non-surgical management approaches.

## 1. Introduction

The management of locally advanced rectal cancer has evolved significantly over the past few decades, primarily due to advancements in surgical techniques, radiotherapy, and systemic therapies. The cornerstone of treatment for stage II and III rectal cancer traditionally involves neoadjuvant chemoradiotherapy (nCRT) followed by surgical resection, typically total mesorectal excision (TME). This approach aims to reduce tumor size and improve surgical outcomes, while also attempting to preserve organ function [1,2]. Neoadjuvant radiochemotherapy has been a standard treatment modality, shown to decrease local recurrence rates and improve overall survival. Previous studies have demonstrated a significant reduction in local recurrence rates from 47% to as low as 6% following nCRT [3]. The introduction of Total Neoadjuvant Therapy (TNT), which includes the addition of full systemic chemotherapy to the neoadjuvant regimen, represents a further evolution in treatment strategy. This approach aims to address not only the primary tumor but also micrometastatic disease, potentially increasing the rates of pathological complete response (pCR) and improving survival outcomes [4]. Recent trials such as RAPIDO and PRODIGE23 have reported pCR rates approaching 28%, highlighting the efficacy of TNT in achieving significant tumor regression [5,6]. Furthermore, therapies with immune checkpoint inhibitors, especially in a neoadjuvant setup, also demonstrated excellent pathologic responses in microsatellite unstable rectal cancer with high mutational burden [7]. The achievement of a pCR—where no viable cancer cells are detected histologically post-treatment—is a particularly favorable outcome, associated with improved long-term survival and reduced need for invasive surgery [3]. The concept of “Watch and Wait” (W&W) has emerged as a viable strategy for patients who achieve a clinically complete response without evidence of residual disease on imaging and endoscopic evaluation. This approach, pioneered by Habr-Gama and colleagues, avoids surgery and associated morbidities in favor of close surveillance [8]. Long-term data suggest that survival outcomes in patients managed with a W&W approach can be comparable to those who undergo surgery, provided that rigorous monitoring and patient selection criteria are met [9]. Accurately predicting which patients will achieve pCR is critical for the success of conservative management strategies. Traditional diagnostic tools can be insufficiently sensitive or specific in predicting pCR [10,11]. Recent advancements have focused on integrating multiple diagnostic modalities to improve the predictive accuracy of treatment response [12]. One of the most promising advancements in disease monitoring is the use of liquid biopsy (LBx). LBx involves the analysis of circulating tumor DNA (ctDNA) in the bloodstream, which can provide real-time, non-invasive insights into tumor dynamics [13]. LBx has the potential to detect minimal residual disease (MRD) with high sensitivity and specificity, even when traditional imaging and histopathology fail to show evidence of residual disease [14].

Therefore, we aimed to monitor patients with locally advanced rectal cancer who are eligible for nCRT and undergo standardized clinical management using longitudinal LBx. We conducted the NEORECT study to quantify informative mutations in ctDNA before nCRT and their dynamics during nCRT and to correlate these results with imaging according to standard treatment, tumor markers, and histological remission status after surgery. We hypothesized that a multimodal assessment including the currently established parameters and LBx would enable prediction of pCR after nCRT and thus have the potential to influence the decision for a W&W approach in the long term.

## 2. Materials and Methods

### 2.1. Patients and Trial Oversight

We performed a prospective non-interventional single-center trial on rectal cancer patients enrolled between December 2017 and September 2019 at the University Hospital of Augsburg (NEORECT). It was approved by the Ethics Committee of LMU Munich (ethical vote number 17-586). The trial was conducted in concordance with the Declaration of Helsinki and the ICH GCP guidelines. Patients with stage II/III rectal cancer undergoing nCRT were eligible and gave written informed consent before trial inclusion.

Diagnosis and nCRT were performed according to standardized procedures as detailed by the current guidelines. Initial diagnostics consisted of rigid rectoscopy including deep biopsies and staging by CT and MRT scans. Mutation profiles in biopsies were determined using the AmpliSeqCancer HotSpot Panel for Illumina^®^ on Illumina MiSeq (Illumina, San Diego, CA, USA). NCRT was carried out according to the “Sauer” protocol [3] and all radiological, endoscopic, pathological, and laboratory tests in this study were obtained exclusively in the context of routine diagnostic procedures. Peripheral blood samples for LBx were collected at the beginning (V1), during (V2) and on the last day (V3) of nCRT and before surgery (V4) concurrently with re-staging (6–8 weeks after nCRT). An overview of the trial design is shown in Appendix A. After surgery, a detailed histopathological examination of the surgical specimens was performed.

### 2.2. Clinical Assessment for Response Evaluation

Radiographic assessment and response evaluation of CT and MRT scans were conducted by the Department of Radiology at the University Hospital of Augsburg. A complete response to nCRT was defined by either no visible residual tumor or complete fibrosis/granulation of residual tumor volume. Partial responses included subgroups of good, moderate, and poor responses, depending on the proportional reduction in tumor size to baseline and/or proportional transformation of residual tumor volume to fibrosis/granulation.

Rectoscopic response was reported based on differences between pretherapeutic and preoperative assessment in terms of size, scarring, and observation of surrounding tissue resulting in categorization as good, moderate, or no response.

Carcinoembryonic antigens (CEAs) above 3.8 ng/mL were defined as elevated according to the local standard.

Surgical specimens were evaluated by pathologists from the Institute of Pathology and described based on the TNM classification for rectal cancer. Based on histopathological examination of the specimen after TME, patient response to nCRT was classified as pCR, subtotal remission (SR), or no pCR. Pathological assessment (Dworak scoring) was defined as ground truth regarding response to overall treatment and used for correlation analysis with other modalities and LBx in this study.

### 2.3. Plasma Sample Collection and cfDNA Isolation

Peripheral blood (4 × 9 mL in EDTA tubes) was obtained at four timepoints. Plasma preparation was performed within two hours. Plasma was isolated with two centrifugation steps at 2000× *g* for 10 min each. If not instantly processed, plasma was stored at −80 °C. CfDNA was isolated from 4 ml plasma using the Maxwell ^®^ RSC LV ccfDNA Kit (Promega, Madison, WI, USA, custom AX1115), eluted in 60 µL nuclease-free water, and quantified fluorometrically (Quantus, Promgea, Madison, WI, USA).

### 2.4. CtDNA Quantification

Informative genetic mutations were identified from initial biopsies by NGS as described above and used for targeted tracking of ctDNA by dPCR (QuantStudio 3D Digital PCR System, Applied Biosystems, Foster City, CA, USA). All dPCRs were run in duplicates and the mean was documented as the percentage of mutated alleles. Here, a detection sensitivity of 0.1% was determined (Appendix A) with values under this limit defined as not detectable. The results are reported as genome equivalents per ml plasma (GE/mL plasma).

### 2.5. Statistical Analyses

Statistical analyses were performed using the R Statistical Computing Environment Version 4.3.0. Comparisons between the two groups were conducted using the non-parametric Wilcoxon test. Pairwise correlations were performed based on Pearson correlation with *p*-values generated using the R package (Version 4.3.0 ) Hmisc and visualized with the package corrplot. Disease-free survival (DFS) was analyzed based on the log-rank test using the R package survminer and visualized by the Kaplan–Meier method. *p*-values below 0.05 were considered statistically significant.

## 3. Results

### 3.1. Patient Population and Tumor Characteristics

The NEORECT trial recruited a total of 40 participants who received standard nCRT over a period of two years at the University Hospital of Augsburg. An overview of basic patient and tumor characteristics is shown in Appendix A. The median age was 68 years (range: 37–87) and 73% were male. The median minimum distance of the tumor ab ano was 7 cm. Nine patients (23%) were staged cT4, and cT3 was diagnosed most frequently (73%). Six patients (15%) had an oligometastatic disease with single distant metastases (four in the liver, one in the lungs, and one in both the liver and lungs). The majority was histopathologically classified as G2 (75%), and eight individuals had no initial grading.

Five of the forty participants stopped the protocol at an early stage, seven had an external TME (no final staging, nor LBx, nor remission status available), and two missed blood drawing before surgery (V4), resulting in twenty-six individuals with a complete sample set and dataset (Figure 1). Four patients lacked an informative driver mutation in primary tissue and no assay for dPCR was available for four cases. A total of 18 patients were eligible for ctDNA tracing.

Panel sequencing of primary biopsies was performed for the identification of traceable mutations for the dPCR assay (Appendix A). Primary biopsy was available for three individuals. One had no primary material available at the time of diagnosis but was analyzed molecularly by NGS after resection. Somatic mutations in 21 different genes were detected ranging from one to nine mutations per patient. From the mutation patterns detected by NGS, a single aberration was selected as a genetic marker for ctDNA analysis based on the highest possible VAF to capture the largest clonal portion of the tumor and on the availability of a technically reliable probe for dPCR testing. Eleven of the eligible individuals were traced by different *KRAS* mutations. The detected *NRAS* and *BRAF* mutations were used to track ctDNA for two patients. *PIK3CA* mutations were detected in three participants, two of whom used the same assay at the p.E545K hotspot. Although *TP53* mutations were most frequently detected in our cohort (62%), they were only used for dPCR-based tracking in two cases.

### 3.2. Distinct ctDNA Dynamics During nCRT

Absolute cfDNA levels and ctDNA dynamics were analyzed at the individual patient level. Three main patterns were observed (Figure 2). Patients defined as not detectable are characterized by ctDNA absence at any timepoint during or after therapy (A). Patients with a positive or negative change in ctDNA quantity before surgery compared to any timepoint during therapy were classified as ctDNA increase (B) or decrease (C), respectively. Four of the five individuals in group A had *RAS* mutations (three *KRAS* and one *NRAS* mutation) in the respective biopsies ranging from 3% to 36% and one patient had a *PIK3CA* mutation. In this group, cfDNA levels ranged from 742 to 6894 GE/mL plasma. It should be noted that P40 was traced by a *KRAS* mutation detected after resection so the alteration may have arisen at a later stage of the disease in a subclone (3% VAFs). The biggest group with 10 individuals (B) was defined by an increment in ctDNA in the interval during nCRT and surgery. Three patients showed the lowest ctDNA level already at V2 while the majority reached the minimum by the end of nCRT (V3). Interestingly, in contrast to the observed steady decline to the lowest level at V3, two individuals (P21 and P26) showed increased ctDNA levels during therapy (V2) before dropping to a lower ctDNA level at V3. Patient P36 lacked a ctDNA value at V1 because the dPCR could not be analyzed, but the increase between V3 and V4 allowed classification into this group. CfDNA quantity varied within this group from 636 to 4468 GE/mL plasma. The third group (C) is characterized by constant ctDNA decrement. Two of these patients had ctDNA detectable only at the beginning of nCRT (V1) and one had positive ctDNA values until V2. At the end of the therapy, no ctDNA could be detected and remained negative until surgery (V4). This group included the two individuals with the highest quantity of ctDNA at baseline (P03: 83.7 and P15: 28.6 GE/ml plasma). The total cfDNA amounts were the lowest in this group ranging from 689 to 2027 GE/mL plasma.

### 3.3. Association of ctDNA with Pathological Response to Therapy and Multimodal Clinical Diagnostics

First, the utility of ctDNA detection as a predictor of response to therapy based on remission status and Dworak score was evaluated. Both the ctDNA group and ctDNA status before surgery as a single timepoint were considered. Of the eighteen patients eligible for ctDNA tracing in our cohort, only one was classified as a pCR after treatment (P01) and one as SR after TME (P04). P05, P37, and P40 also had no detectable ctDNA over the entire assessment period but were pathologically classified as no pCR. Individuals from the other two groups with increasing or decreasing ctDNA also showed no pCR. However, statistical analysis based on the Wilcoxon rank-sum test confirms a significant correlation between the ctDNA status before surgery as a single timepoint and the Dworak score (W = 18, *p* = 0.01528). Further clinical diagnostic modalities were taken into consideration and correlated with pathohistological classification after TME (Figure 3A). PCR status (P01) was consistent with MRT observations before surgery and CEA values within the physiological range. However, the individual was ranked to show “no response” in the rectoscopic assessment. This pattern is similar in the patient described with an SR (P04) despite having elevated CEA values. Of the 13 patients with detectable ctDNA, 38,5% of the resctoscopic examinations described a good response to therapy, and 54% had physiological CEA values. MRT assessment reported a poor response to treatment in only two cases. Remarkably, although three (P15, P21, P33) showed good responses in all pre-surgical assessments, remaining tumor cells could be histologically detected easily in the resected specimen (Dworak 2). A correlation analysis of all parameters to recognize potential in-between linkage is shown in Figure 3B. As expected, the highest correlations can be observed between the pathological assessed remission status, Dworak score, and ypT. MRT response assessment also correlates well with the remission status and ypT but poorly with the Dworak scoring (correlation coefficient: 0.37). The observed rectoscopic response and pre-surgical CEA values depict only low correlations to the other parameters. Within the LBx analysis, a negative correlation to the post-surgical parameters (remission status, Dworak, and ypT) can be noted. Taken together, although trends are recognizable, only a weak correlation between the pre-surgical parameters can be reported.

### 3.4. Testing of a Composite Approach of Diagnostic Modalities Before Surgery

As the pre-surgical assessments appeared to be only moderately correlated, a scoring system combining these parameters was tested to provide a more refined clinical prediction of tumor response to therapy. One point was assigned for a reported complete/good response in MRT and rectoscopy, physiological CEA values (0–3.8 ng/mL), and ctDNA negativity before surgery. All other classifications (worse response), elevated CEA values, and ctDNA positivity at V4 were ranked with zero points. This results in a scoring system ranging from zero to four (the higher the score, the better the estimated pathological response) (Figure 4A).

Only one patient showed a complete/good response to treatment in all categories (P15) although this patient was post-surgically described as Dworak 2. Two of the four individuals with three points had a positive ctDNA value before surgery. Within the four and seven participants with two and one points, respectively, the distribution varies between the categories. In total, two individuals were assigned zero points. In two cases, rectoscopy was not assessable, and two CEA values were not available. These missing instances were regarded as zero (overall distribution: Score 0: n = 2; Score 1: n = 7; Score 2: n = 4; Score 3: n = 4; Score 4: n = 1). The arbitrary composite score was put in the context of the Dworak regression score (Figure 4B). Overall, no significant correlation could be observed between the composite scoring based on pre-surgical parameters and pathologically categorized Dworak scores after TME (Pearson’s Chi² = 8.153846; d.f. = 9; *p* = 0.5187198).

### 3.5. ctDNA Dynamics in a Setting of Metastatic Progression

In one case (P06), clinical assessment before TME showed the appearance of previously unrecognized metastases in the liver so a “liver-first approach” was followed. LBx analyses were extended over two more samplings (V5 and V6) until the date of resection of the primary tumor (Figure 5). Here, ctDNA was constantly detectable during nCRT (V1: 12.7; V2: 13; V3: 8 GE/mL plasma). LBx before TME (V4) showed a tenfold increase in ctDNA (88.5 GE/mL plasma). As the treating physicians decided to follow the “liver-first” protocol, the participant was further observed in the context of the trial. Sequencing of the resected liver metastasis showed a VAF of 84% of the initially detected *TP53* mutation. The additional sample obtained two weeks after liver surgery (V5) showed complete ctDNA clearance. CtDNA remained negative until TME (V6). Final sequencing of the primary tumor after resection yielded a VAF of 43% and the individual was pathologically classified as no pCR with a Dworak score of two.

### 3.6. Association of ctDNA Analysis and Disease-Free Survival

Independently of the predictive value of the investigated parameters (including ctDNA status) for the pathological remission status before TME, an analysis of the impact on disease-free survival (DFS) was conducted (Figure 6). DFS was defined as no recurrence of primary tumor or distant metastases at the time of last contact (between 1 and 52 months after surgery, median follow-up: 29 months). DFS depending on the pathological remission status (grouped pCR and SR vs. no PCR) determined after TME (A) was 100% versus 72% after 24 months and 100% versus 64% after 36 months, respectively. When stratified by ctDNA dynamics grouping (no ctDNA vs. increment vs. decrement; B), no significant DFS differences (log-rank *p*-value = 0.98) could be detected. DFS was 67% after one and three months for the groups with ctDNA decrement and no ctDNA detectable, respectively. In contrast, in the group of ctDNA increment, a DFS of 65% was reported after 25 months. Depending on the ctDNA status at baseline (V1, positive versus negative), DFS was 86% versus 57% for ctDNA-negative and -positive individuals, respectively (C). After 36 months, the DFS of the group with positive ctDNA at baseline stayed constant at 57% versus 69% for the ctDNA negative group but no significant differences could be detected (log-rank *p*-value= 0.53). This also applies when stratified by ctDNA before TME (V4) as a single timepoint (D, log-rank *p*-value = 0.9). Here, depending on ctDNA detectability before TME (negative vs. positive), DFS was 70% versus 78% after 24 months and 70% versus 65% after 36 months, respectively.

## 4. Discussion

### 4.1. Liquid Biopsy in the Context of the Sauer Protocol for Locally Advanced Rectal Cancer

In the past decades, great advances have been made in the treatment of rectal cancer [15]. In fact, the implementation of combined chemo- and radiotherapy before surgery has led to up to 30% of patients showing a pCR after treatment [9]. Although W&W approaches have worked well within this population, there is an unmet need for predictive factors to detect these individuals reliably before TME [16]. Here, we used a dPCR-based targeted LBx approach to investigate the predictive role of ctDNA in patients with locally advanced rectal cancer. Our aim was to examine the ability of LBx for outcome prediction under nCRT for the identification of patients eligible for a W&W approach.

The 40 participants enrolled in the NEORECT trial underwent the well-established Sauer protocol consisting of combinational chemo- and radiotherapy before TME. Despite the relatively low number of individuals in our study, our cohort reflected key features of patients with locally advanced rectal cancer as described in the literature (e.g., mutational landscape) [17,18]. The rate of pCR was 14% and is therefore slightly lower than the described 20% to 30% [9]. The 12 patients with missing information from external surgeries could possibly compensate for the remaining discrepancy.

We were able to detect ctDNA before starting nCRT (V1, baseline) in over 60% of the participants. As we included locally advanced tumors, ctDNA before treatment may originate from foci of necrotic or apoptotic cell death which are described to occur spontaneously in advanced solid malignancies [19]. The differences in ctDNA positivity at baseline could be due to differences in tumor burden or other factors such as vascularization and the tumor environment [20]. Interestingly, both patients with pCR/SR had negative ctDNA results before treatment initiation.

By the end of nCRT (V3), ctDNA was detected in only a small proportion of patients, which reflects the expected tumor shrinkage. This dynamic was previously described not only for ctDNA but also for CTCs in the context of rectal cancer [13]. However, at this timepoint, ctDNA did not reliably discriminate between good and poor responders. This was also observed during LBx assessment directly before surgery (V4). As there are several treatment-free weeks before TME, ctDNA at this timepoint is of special interest. Here, again, over 50% of patients showed ctDNA positivity.

A correlation between V4 ctDNA status and postoperative Dworak scores was confirmed in our study. However, not all V4 ctDNA-negative patients achieved pCR. This discrepancy was already reported by Tie and Cohen and colleagues in a 2019 study. They also concluded that ctDNA analysis within a short interval after nCRT is not sufficient to discriminate patients eligible for a W&W approach [21].

In our study, the preoperative LBx analysis in one participant was highly remarkable, as it was ten times higher than at the end of therapy. Here, new metastases occurred in the treatment-free interval. Sequencing results showed that the new liver metastases originated from cells carrying the detected mutation. This case underscores the importance of ctDNA in reflecting overall tumor burden and indicating early disease progression.

Taken together, our study shows that, while preoperative ctDNA negativity at a single timepoint may indicate a clear response (pCR/SR), it does not appear to be sufficient to reliably distinguish responders from non-responders—the basis for a W&W approach. These findings are in line with previous studies [21,22].

We also considered ctDNA dynamics based on the assumption that the decrement in ctDNA quantity during nCRT is presumably due to tumor shrinkage. In a study in 2020, Murahashi and colleagues described two groups of ctDNA dynamics during preoperative therapy of locally advanced rectal cancer. It is noteworthy that only individuals with positive ctDNA results in at least one measurement were included hence omitting cases with constant negative results. These authors examined ctDNA at baseline and after preoperative treatment defined as before surgery and found a significant association between response to therapy and ctDNA changes. Good responders showed decreasing ctDNA dynamics and non-responders showed increasing ctDNA dynamics, respectively [22]. This is comparable to the classification defined in our study. Patients with excellent responses (pCR and SR) had undetectable ctDNA at all timepoints, though this was not an exclusive feature of good responders. This observation is supported by the results from Carpinetti and colleagues, who described negative ctDNA levels in a patient with pCR but, unfortunately, also in patients presenting incomplete responses even with significant tumor regression [23]. Overall, analysis of ctDNA dynamics in our trial did not reliably distinguish patient responses to therapy.

### 4.2. Prognostic Value of Liquid Biopsy Incorporating Multimodal Aspects and Disease-Free Survival

As LBx alone does not appear to be sufficient to reliably identify patients with pCR for a W&W approach, we hypothesized that building a scoring system with other routinely obtained modalities might show a cumulative effect. Previous studies have already examined LBx analysis combined with other modalities. In a study in 2021, Osumi et al. evaluated the relationship between CEA values and ctDNA in 110 individuals with rectal cancer. Similarly to our results, they reported a low correlation. Furthermore, they described both parameters to be affected by tumor volume with an increased number of false negative results in smaller tumor cases [24]. Interestingly, the preoperative CEA values of P06, who had new liver metastases before the planned TME, were in the physiologic range. In this case, there is a large discrepancy between the two variables, with LBx giving a more accurate picture of the actual disease state. Another study of 2020 also aimed to predict pathological response after nCRT using ctDNA. Here, a positive predictive value of ctDNA combined with endoscopic findings was described [22]. However, rectoscopy results are not always assessable. Likewise, Wang and colleagues explored the value of ctDNA in combination with MRT and also described an improvement in the predictive performance in a combined model compared to the individual information [25].

In the NEORECT trial, we proposed an equally weighted scoring system consisting of all four parameters: CEA levels, MRT response, rectoscopic response, and ctDNA. For instance, both the individual with a Dworak score of 4 (total pCR) and the patient with the worst treatment response (Dworak 1) were assigned three points in our multimodal scoring. Therefore, although trends are recognizable, it is not possible to reliably discriminate between patients with good or poor responses to nCRT.

Previous clinical trials indicate that patients following a W&W approach after nCRT show excellent outcomes after pCR without TME with a DFS of 86% [8]. In NEORECT, neither dynamics nor individual timepoint measurements of ctDNA significantly predicted DFS. In this context, previous studies have shown the power of post-surgical ctDNA as a marker for MRD and as a predictor for recurrence-free survival [14,22,26]. Investigation of ctDNA after, rather than before, TME appears to be the more precise tool for DFS analysis.

In conclusion, the NEORECT trial provides an interesting pilot platform to investigate the behavior of ctDNA in the neoadjuvant setting and generates information on how personalized ctDNA monitoring can complement imaging and clinical assessment of tumor response. However, as the ctDNA-related effects were not sufficient to reliably identify excellent responders and possibly replace TME with a W&W approach in these patients, this trial was not successful in defining a clear role for ctDNA monitoring in the therapy management of locally advanced rectal cancer.

### 4.3. Strengths and Limitations of Our Study

Several features of our study design stand out. We used an “informed approach”, which refers to the identification of variants from primary tumor tissue. This approach has particular advantages when tumor-associated variants are to be reliably distinguished from alterations in clonally expanded hematopoietic stem cells. The major advantages of testing the respective mutations using dPCR are the high sensitivity, the low cost per sample, and the short turnaround time.

Previous studies have explored the correlation between ctDNA and single clinical factors [22,27]. Thus, we aimed for a multimodal concept and included the key available clinical parameters routinely collected prior to TME to provide the best possible overall assessment of tumor response. To the best of our knowledge, this is the first study to incorporate all these variables and evaluate a multimodal scoring system.

However, there are also some limitations. Firstly, the number of recruited patients in our pilot trial was low and the drop-out rate was high. A sufficiently powerful multicenter study would help to generate definitive, statistically significant answers to the questions addressed with a sufficient number of participants. Furthermore, cooling during centrifugation might improve cfDNA yields for subsequent analyses. Also, the ability to detect ctDNA via dPCR was limited in many participants. Although personalized assays could be developed, this may not be feasible in everyday care. Another drawback of the targeted dPCR approach is that potentially relevant clones are not sufficiently covered by tracking only one clonal marker, thus neglecting information on inter- and intratumoral heterogeneity. Therefore, metastases from subclones harboring other genetic profiles would falsely have no impact on the ctDNA analysis, and information about longitudinal clonal evolution is lost. Finally, although many modalities were included in our study, many conceivable confounding factors such as gender, distance from the anal verge, tumor size, and concomitant diseases were not considered in the analyses. Further studies with comprehensive clinical data and larger numbers of patients are needed to additionally consider and statistically evaluate these parameters.

## 5. Conclusions

In conclusion, the treatment of locally advanced rectal cancer is rapidly evolving, with an improvement in neoadjuvant strategies such as the establishment of TNT playing a critical role in patient outcomes. The role of LBx as a method of detecting MRD and thus predicting pCR appears to have potential, but the available data are not yet fully conclusive. It is likely that the integration of different preoperative parameters will be necessary to generate optimal predictive results and make treatment more effective and less invasive.

## Figures and Tables

**Figure 1 cancers-16-04173-f001:**
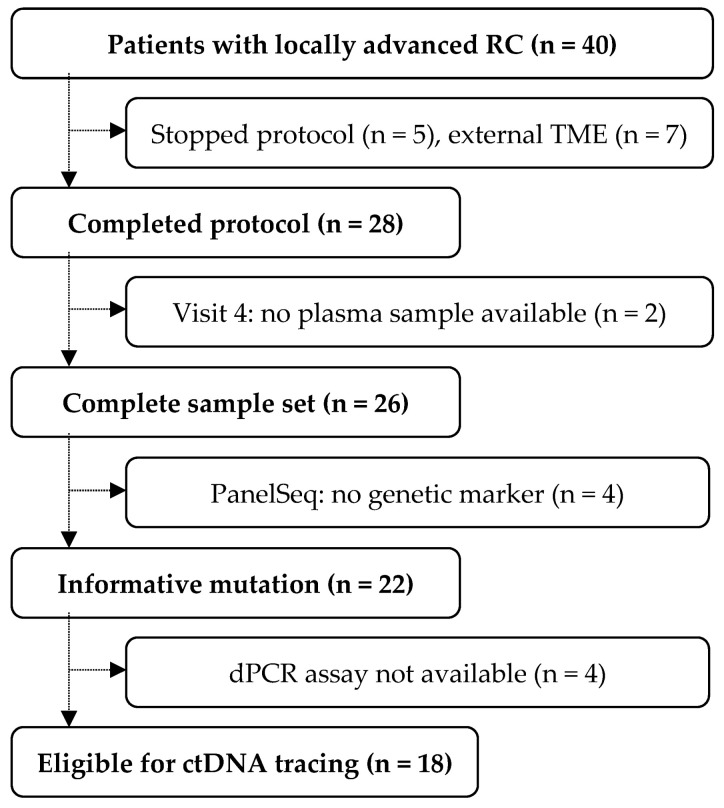
Patient disposition in the NEORECT trial. Of the 40 patients intended to be treated, 26 had a complete dataset after surgery. Due to technical limitations, 18 individuals were eligible for ctDNA tracing in plasma samples.

**Figure 2 cancers-16-04173-f002:**
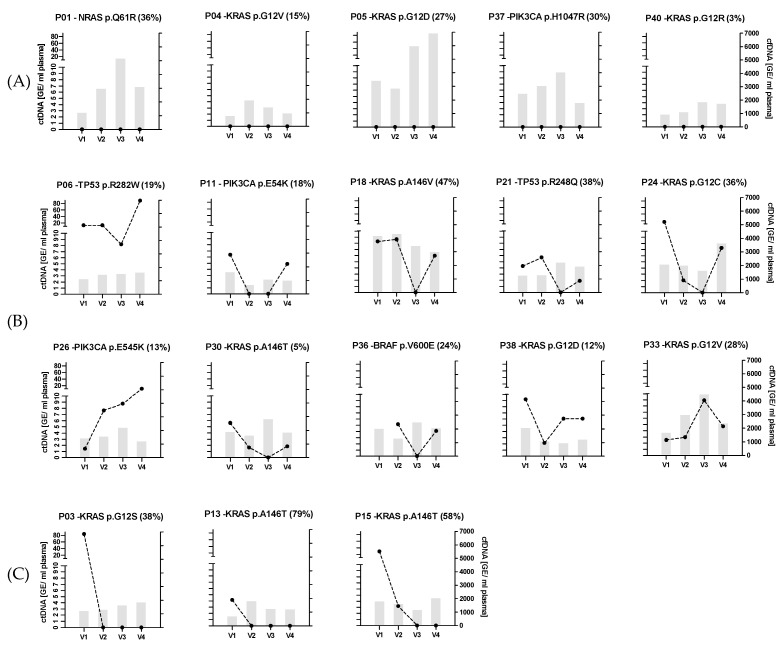
CtDNA and cfDNA dynamics during treatment of the 18 patients of the NEORECT trial. Dot lines depict ctDNA dynamics as genome equivalents per milliliter plasma (GE/mL plasma, left y-axis) and gray bars show cfDNA as GE/mL plasma (right y-axis). Individual headlines describe the patient ID of the NEORECT trial and its mutated gene and hotspot traced by dPCR as well as the VAFs detected by NGS from the initial biopsy in brackets. Three groups can be defined based on ctDNA dynamics: Undetectable ctDNA at any timepoint (**A**), increment of ctDNA towards V4 compared at any timepoint during nCRT (**B**), and overall ctDNA decreasing over the course of therapy (**C**).

**Figure 3 cancers-16-04173-f003:**
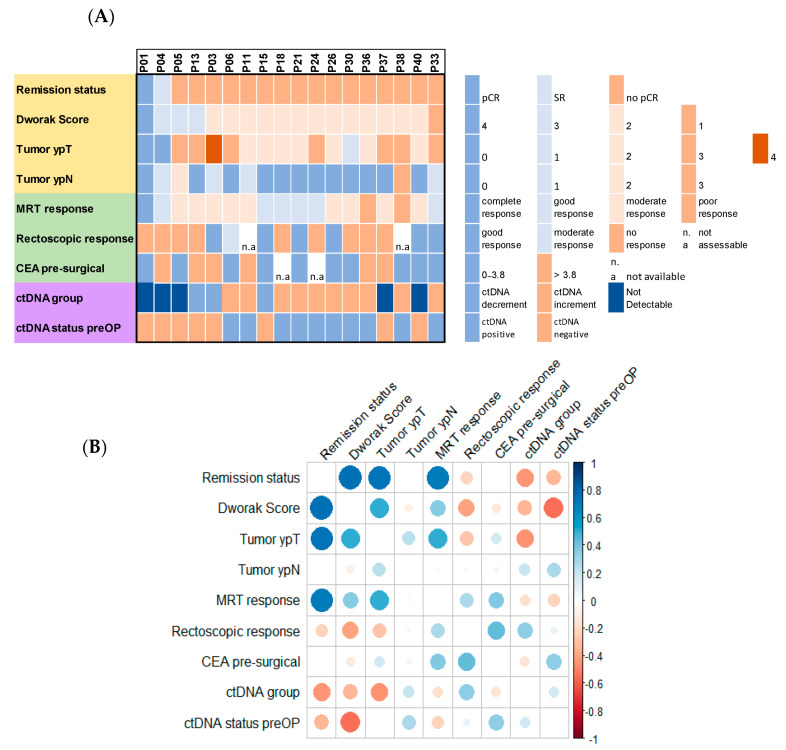
Multimodal disease evaluation before and after surgery based on different clinical and pathological parameters and LBx (n = 18) (**A**) Yellow: pathological assessment after surgery; green: clinical assessment before surgery; purple: LBx assessment; pCR: pathological complete remission; SR: subtotal remission; CEA: carcinoembryonic antigen. (**B**) pairwise Pearson correlation of all parameters. Blue depicts a positive correlation and red a negative correlation, respectively. No significant coefficients are shown as blank.

**Figure 4 cancers-16-04173-f004:**
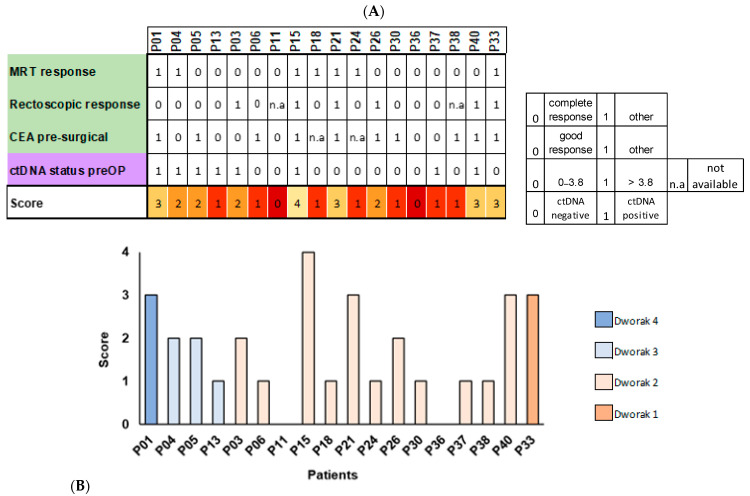
Composite score of clinical diagnostic modalities and ctDNA before surgery (V4). (**A**) Forming a pre-surgical scoring based on clinical parameters assessed in the context of clinical routine diagnostics and ctDNA status at V4 (NEORECT scoring). CEA: carcinoembryonic antigen; n.a: not assessable/not available; preOP: before surgery. (**B**) NEORECT scoring of individual participants. Coloring is dependent on pathologically classified Dworak scoring after surgery as depicted for Dworak 4 to 1.

**Figure 5 cancers-16-04173-f005:**
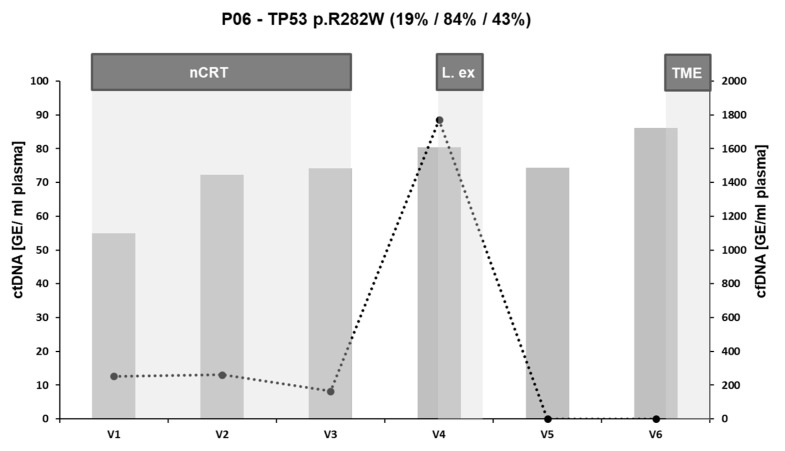
Dynamics of LBx (ct and cfDNA) in a patient with metastatic recurrence during nCRT and a liver-first approach. Both ctDNA (dotted line) and cfDNA (gray bars) are described as genome equivalents per milliliter plasma (GE/mL plasma). The headline describes the patient ID and the mutated gene and hotspot traced by dPCR as well as the VAFs detected by NGS in the respective tissue in brackets (primary tissue/liver metastasis/resected specimen). nCRT: neoadjuvant chemoradio therapy; L. ex: excision of liver metastases; TME: total mesorectal excision.

**Figure 6 cancers-16-04173-f006:**
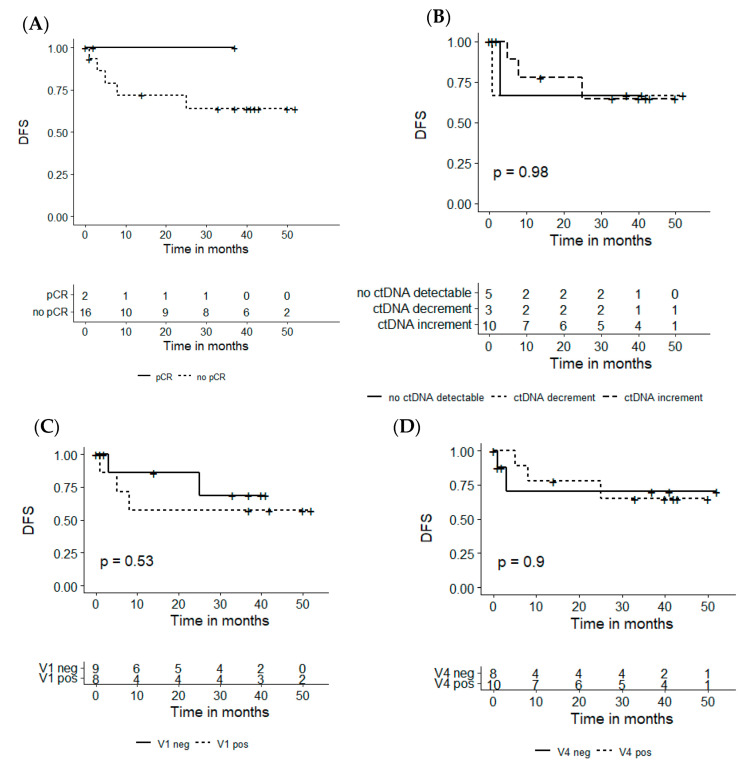
Disease-free survival (DFS) stratified by histological response and ctDNA measurements. Patients were stratified (**A**) by histopathological response status, (**B**) by ctDNA dynamics, (**C**) depending on V1 ctDNA (even distribution), and (**D**) depending on V4 ctDNA. Statistical significance was tested based on the log-rank method. *p*-Values under 0.05 are defined as statistically significant.

## Data Availability

The data presented in this study are available upon request from the corresponding authors due to privacy and ethical reasons.

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
