# Peer review of "Circulating Tumor DNA for Prediction of Complete Pathological Response to Neoadjuvant Radiochemotherapy in Locally Advanced Rectal Cancer (NEORECT Trial)"

_cancers, 2024, doi:10.3390/cancers16244173_

Round 1

Reviewer 1 Report

Comments and Suggestions for Authors

This reseach discusses on whether ctDNA alone or in combination with the other well-established biomarkers can contribute to the prediction of response to neoadjuvant therapy in locally advanced rectal cancer. The study is well designed, detailed information was gathered for each patient, and the manuscript is well written. However, the cohort size is small as the authors described in their manuscript, and the number of pCR and SR is too small, which might be the reasons for absence of any positive findings on the role of ct DNA. However, multimodal approach incorporating the other biomarkers seems to be good approach. Though scoring system did not provide any good information on the response prediction, the other appoaches such as decision tree might produce different results because model fitness to the data is higher.

Overall though the cohort size and composition has fundamental limitation, this research can provid interesting results to the readers in this journal.

Author Response

Comment 1: This reseach discusses on whether ctDNA alone or in combination with the other well-established biomarkers can contribute to the prediction of response to neoadjuvant therapy in locally advanced rectal cancer. The study is well designed, detailed information was gathered for each patient, and the manuscript is well written. However, the cohort size is small as the authors described in their manuscript, and the number of pCR and SR is too small, which might be the reasons for absence of any positive findings on the role of ctDNA. However, multimodal approach incorporating the other biomarkers seems to be good approach. Though scoring system did not provide any good information on the response prediction, the other appoaches such as decision tree might produce different results because model fitness to the data is higher.
Overall though the cohort size and composition has fundamental limitation, this research can provid interesting results to the readers in this journal.

Response: We thank reviewer 1 for his comprehensive assessment and the appreciation of our work.

Reviewer 2 Report

Comments and Suggestions for Authors

Dear Editor,

I have reviewed the work of Mögele T, et al. Titled: “Circulating tumor DNA for prediction of complete pathological response to neoadjuvant radiochemotherapy (CRT) in locally advanced rectal cancer (NEORECT trial)“. This is a prospective single center study of 18 patients with locally advanced rectal cancer treated with neoadjuvant chemoradiotherapy followed by total mesorectal. The main objective was to analyze ctDNA changes during nCRT identifying individuals reaching pCR. Three distinct ctDNA patterns were obtained and undetectable DNA was associated in good responders and ctDNA increase was associated with new metastases. Study falls showing correlation to pathological complete response (pCR) or long-term prognosis. The study is well writing and could be interesting to the journal. Below I point out my comments:

Materials and Methods:

Plasma sample collection and cfDNA isolation

Peripheral blood (4 x 9 ml in EDTA-tubes) was obtained at four timepoints. Plasma preparation was performed within two hours. Plasma was isolated with two centrifugation steps at 2000 × g for 10 min each.

Q1: The use of “cool” centrifugation can improve the cfDNA, why the authors did not use this procedure?.

ctDNA quantification

Informative genetic mutations were identified from initial biopsies by NGS as described above and used for targeted tracking of ctDNA by dPCR.

Q2: dPCR is an excellent option to follow-up patients with hot spot mutations. However, during the follow-up could appear new mutations, therefore the cfDNA quantification or ctDNA tumoral fraction can increase the possibility to obtain more util information. Why the authors did not use total ctDNA quantification?. It can be correlated with the specific mutation?. The analyses of both quantifications should be include.

Author Response

Comment 1: I have reviewed the work of Mögele T, et al. Titled: “Circulating tumor DNA for prediction of complete pathological response to neoadjuvant radiochemotherapy (CRT) in locally advanced rectal cancer (NEORECT trial)“. This is a prospective single center study of 18 patients with locally advanced rectal cancer treated with neoadjuvant chemoradiotherapy followed by total mesorectal. The main objective was to analyze ctDNA changes during nCRT identifying individuals reaching pCR. Three distinct ctDNA patterns were obtained and undetectable DNA was associated in good responders and ctDNA increase was associated with new metastases. Study falls showing correlation to pathological complete response (pCR) or long-term prognosis. The study is well writing and could be interesting to the journal. Below I point out my comments:
Materials and Methods:
Plasma sample collection and cfDNA isolation
Peripheral blood (4 x 9 ml in EDTA-tubes) was obtained at four timepoints. Plasma preparation was performed within two hours. Plasma was isolated with two centrifugation steps at 2000 × g for 10 min each. Q1: The use of “cool” centrifugation can improve the cfDNA, why the authors did not use this procedure?.

Response 1: We thank reviewer 2 for this relevant comment. To our knowledge the concept of cooling during centrifugation for cfDNA isolation was not well established when we conducted our trial and other studies found in the literature such as Tie et al., 2016, didn´t comment on cooling samples. Furthermore, our established protocol has been successful in over 600 plasma samples isolated in the ALPS study (Sommer and Schmutz et al., 2024; https://doi.org/10.1515/labmed-2023-0156). However, we will include this relevant remark in our discussion.

Comment 2: Informative genetic mutations were identified from initial biopsies by NGS as described above and used for targeted tracking of ctDNA by dPCR.
Q2: dPCR is an excellent option to follow-up patients with hot spot mutations. However, during the follow-up could appear new mutations, therefore the cfDNA quantification or ctDNA tumoral fraction can increase the possibility to obtain more util information. Why the authors did not use total ctDNA quantification?. It can be correlated with the specific mutation?. The analyses of both quantifications should be include.

Response 2: Methods like panel sequencing to address tumors´ mutational landscape more comprehensively or shallowWGS (for CNV detection) are conceivable. We are currently incorporating this aspects into our trials. However, for this trial we focused on an easy and feasible assay which has low hurdles to be transferred into a routine care setting. Furthermore, for NEORECT, in the context of rectal cancer, we expected RAS and BRAF mutated clones to be the major clones in most of the cases. This is also likely to be stable over the neoadjuvant treatment.  However, we cannot formally rule out that other mutations/clones may arise. We have commented this topic as a limitation in our discussion.

Reviewer 3 Report

Comments and Suggestions for Authors

For locally advanced rectal cancer  patients who receive neoadjuvant chemoradiotherapy , there are no reliable indicators to accurately predict pathological complete response (pCR) before surgery. For patients with clinical complete response , a Watch and Wait approach can be adopted to improve quality of life. However, W&W approach may increase the recurrence risk in patients who are judged to be cCR but have minimal residual disease .  In this prospective cohort study, the Authors explored the value of circulating tumor DNA (ctDNA)  in the prediction of pCR  and investigated the utility of ctDNA in risk stratification and prognostic prediction for patients undergoing nCRT and total mesorectal excision (TME).

Thre are some insufficietnt data : is not clear the gender , any correlationship with MR or CAT scan or beetwen distance from the anal verge ( 1cm ore 2, 3, 4?), diameter of the tumor and colonscopy aspect and concomitant diseases as praevious cancer treated were not included in the report.  These aspect could be easily included or at least some of these.

It is of value that the Authors aimed  included the key available clinical parameters  collected prior to TME to provide the best possible over all assessment of tumor response. I agree that this, very probably is  the first study to incorporate all  variables and evaluate with a multimodal scoring system.

Unfortunately, the conclusions of this good study are weak due to the low number of patients and the drop out and also the ability to detect ctDNA via dPCR was limited in many participants.

Even if the  drawback of the targeted dPCR approach is that  some interesting clones are not completely covered by tracking only one  marker, the paper is solid and acceptable.

Comments on the Quality of English Language

English is well written, anyway an assessment from a native speaker could be useful to raise the level

Author Response

Comment 1: For locally advanced rectal cancer  patients who receive neoadjuvant chemoradiotherapy , there are no reliable indicators to accurately predict pathological complete response (pCR) before surgery. For patients with clinical complete response , a Watch and Wait approach can be adopted to improve quality of life. However, W&W approach may increase the recurrence risk in patients who are judged to be cCR but have minimal residual disease .  In this prospective cohort study, the Authors explored the value of circulating tumor DNA (ctDNA)  in the prediction of pCR  and investigated the utility of ctDNA in risk stratification and prognostic prediction for patients undergoing nCRT and total mesorectal excision (TME).
Thre are some insufficietnt data : is not clear the gender , any correlationship with MR or CAT scan or beetwen distance from the anal verge ( 1cm ore 2, 3, 4?), diameter of the tumor and colonscopy aspect and concomitant diseases as praevious cancer treated were not included in the report.  These aspect could be easily included or at least some of these.

Reponse 1: We agree  that incorporation of additional clinical parameters would potentially strengthen the analyses. However, given the limited number of study participants, we might not be able to generate solid correlations/meaningful associations. Gender and distance from anal verge were shortly described in the tumor characteristics section and are shown in the supplemental table. However, given the limited number of study participants and the low number of complete/ subtotal responses, we are not able to conclude on any significant correlations between these parameters. While MRT and CT scans have been incorporated into our multimodal assessment, tumors´ diameters assessed by coloscopy were not specifically included besides the T of the TNM score. Also, concomitant diseases were not assessed in this study and thus cannot be incorporated into our multimodal analysis. We appreciate the reviewers suggestions and agree thtat incorporation of additional/multiple parameters would potentially be relevant. However, due to the limited number of participants, we believe that meaningful associations or interactions between these parameters might be hard to assess. Therefore we incorporated this into our discussion and emphasize the need of bigger studies with a larger number of patients.

Comment 3: It is of value that the Authors aimed  included the key available clinical parameters  collected prior to TME to provide the best possible over all assessment of tumor response. I agree that this, very probably is  the first study to incorporate all  variables and evaluate with a multimodal scoring system.
Unfortunately, the conclusions of this good study are weak due to the low number of patients and the drop out and also the ability to detect ctDNA via dPCR was limited in many participants.
Even if the  drawback of the targeted dPCR approach is that  some interesting clones are not completely covered by tracking only one  marker, the paper is solid and acceptable.

Response 3: We thank the reviewer again for this positive comment. We are convinced that a multimodal assessment rather than a single parameter might be key to identify patients who profit from a watch and wait approach. Thus, we suggest this in our discussion/conclusion as a valuable approach which should be addressed in a larger, sufficiently powered study cohort.

Comment 4: English is well written, anyway an assessment from a native speaker could be useful to raise the level

Response 4: We followed the reviewer’s suggestion and had the languange checked by a native speaker.